# Modulation the Synergistic Effect of Chitosan-Sodium Alginate Nanoparticles with Ca^2+^: Enhancing the Stability of Pickering Emulsion on D-Limonene

**DOI:** 10.3390/foods13040622

**Published:** 2024-02-19

**Authors:** Qian Li, Rui Li, Fanxing Yong, Qiaoli Zhao, Jing Chen, Xing Lin, Ziyu Li, Zhuo Wang, Baojun Xu, Saiyi Zhong

**Affiliations:** 1Guangdong Provincial Key Laboratory of Aquatic Product Processing and Safety, Guangdong Province Engineering Laboratory for Marine Biological Products, Guangdong Provincial Engineering Technology Research Center of Seafood, Guangdong Provincial Science and Technology Innovation Center for Subtropical Fruit and Vegetable Processing, College of Food Science and Technology, Guangdong Ocean University, Zhanjiang 524088, China; qianl1@yeah.net (Q.L.); liruihn@163.com (R.L.); 18035535189@163.com (F.Y.); zql2819557995@163.com (Q.Z.); 15292604455@163.com (J.C.); lxing0206@163.com (X.L.); lzywjp833@163.com (Z.L.); wangzhuo4132@outlook.com (Z.W.); 2Food Science and Technology Programme, Department of Life Sciences, Beijing Normal University-Hong Kong Baptist University United International College, Zhuhai 519087, China; baojunxu@uic.edu.cn; 3Shenzhen Research Institute, Guangdong Ocean University, Shenzhen 518108, China; 4Collaborative Innovation Center of Seafood Deep Processing, Dalian Polytechnic University, Dalian 116034, China

**Keywords:** homogenization speed, thermal stability, ultraviolet stability, antibacterial activity, antioxidant activity

## Abstract

Pickering emulsions (PEs) have been regarded as an effective approach to sustaining and preserving the bioactivities of essential oils. The aim of this research is to prepare a PE stabilized by chitosan/alginate nanoparticles (CS-SA NPs) for the encapsulation and stabilization of D-limonene. In this work, the influence of calcium ions (Ca^2+^) on the morphology and interaction of nanoparticles was studied, and then the preparation technology of CS-SA/Ca^2+^ NPs was optimized. The results showed that the presence of Ca^2+^ reduced the size of the nanoparticles and made them assume a spherical structure. In addition, under the conditions of 0.2 mg/mL CaCl_2_, 0.6 mg/mL SA, and 0.4 mg/mL CS, the CS-SA/Ca^2+^ NPs had the smallest size (274 ± 2.51 nm) and high stability (−49 ± 0.69 mV). Secondly, the PE was prepared by emulsifying D-limonene with CS-SA/Ca^2+^ NPs, and the NP concentrations and homogenization speeds were optimized. The results showed that the small droplet size PE could be prepared with 2 mg/mL NP and a homogenization speed of 20,000 r/min, and it had excellent antibacterial and antioxidant activities. Most importantly, the emulsion showed higher activity, higher resistance to ultraviolet (UV) and a higher temperature than free D-limonene. This research provides a feasible solution for the encapsulation, protection and delivery of essential oils.

## 1. Introduction

D-Limonene, a natural monoterpene hydrocarbon, widely exists in citrus peels such as oranges, lemons, pomelos, and other *Citrus* spp. [1]. As an essential oil, D-limonene has high-value applications in the fields of food, medicine, and the cosmetics industry due to its insecticidal, antioxidant, and anticancer properties [2]. Furthermore, D-limonene has effective resistance to pathogenic bacteria such as *S. aureus*, *Listeria monocytogenes*, *Salmonella enteric* and fungi such as *Penicillium* and *Aspergillus*, which provides a possibility for D-limonene to be utilized as a promising food preservative [3,4]. Unfortunately, the widespread use of D-limonene in the food industry is limited due to its poor stability and solubility [5]. Therefore, it is necessary to overcome the above two problems with D-limonene in order to broaden its application fields. Currently, emulsion-based encapsulation has been widely used as a feasible method to protect D-limonene from light, oxidation, high temperatures, etc., in harsh environments during processing and storage [6].

Generally, the emulsions can be divided into two categories: the traditional emulsions stabilized by small molecules of synthetic surfactants and/or emulsifiers, and the Pickering emulsions (PEs) stabilized by natural colloid particles. PEs exhibit excellent resistance to both Oswald ripening and coalescence and present less toxicity and higher stability than the traditional emulsions [7]. The colloidal particles are composed of polysaccharides, proteins, polyphenols and metal particles; thus, their potential applications in PE have attracted more attention due to their advantages of non-toxicity and environmental friendliness [8,9]. Recently, marine polysaccharides such as chitosan (CS), sodium alginate (SA), carrageenan, and *Ulva fasciata* have also been developed as nanoparticles for further preparation of PEs. Among them, ovalbumin-ferulic acid-κ-carrageenan PE was a successful substitute for butter in bread [10], and *Ulva fasciata*-R-(+)-limonene PE effectively improved the stability of limonene [11]. In addition, CS and SA are the most widely used in Pickering emulsion due to their good biocompatibility.

Produced by the deacetylation of chitin (less than 50% acetylation), CS is the only natural cationic polysaccharide and the second largest sugar reserve after cellulose in nature [12]. Due to its positive charge, CS can form colloidal particles with negatively charged or neutral polymers (e.g., gum Arabic, octenyl succinic anhydride starch, gelatin) under acidic conditions [13,14,15]. Those particles showed good emulsification in PEs of cinnamon essential oil, peppermint oil and astaxanthin [16,17,18]. SA is a linear water-soluble polysaccharide extracted from brown algae that is composed of β-D-mannuronic acid and α-L-guluronic acid [19]. When Ca^2+^ is present, guluronic acids in SA will exchange Na^+^ and react with Ca^2+^, resulting in the cross-linking of guluronic acid fragments to form a three-dimensional network structure [20]. To optimize performance, SA can be blended with other polysaccharide solutions to fabricate complex colloidal particles. Aras Rafiee et al. demonstrated that NPs made of SA and CS have more stability and a smaller particle size than either polymer alone [21]. In particular, the CS-SA NPs were able to effectively encapsulate curcumin, improve its stability, and show excellent emulsifying properties and a good controlled release effect [22]. Moreover, SA NPs are often used to stabilize a variety of PEs, such as β-carotene PE [23], tocotrienol PE [24], resveratrol PE [20], and curcumin PE [25]. Although CS and SA are well-known natural polysaccharides individually, there is limited information about the combination of their complexes as colloidal particles for the preparation of D-Limonene PEs, and the effect of Ca^2+^ on the morphology of CS and SA nanoparticles and the protective effect of nanoparticles on D-limonene are unclear.

Therefore, in this study, a stable D-limonene PE was prepared by attaching chitosan-sodium alginate nanoparticles (CS-SA/Ca^2+^ NPs) to the surface of D-limonene droplets. The effects of CaCl_2_ concentration, SA concentration, CS viscosity and concentration on particle size, polydispersity index (PDI), and potential, as well as the effects of CS-SA/Ca^2+^ NPs concentration and homogeneous speed on the droplet size, zeta potential, and encapsulation efficiency of D-limonene, were investigated. The morphology of CS-SA/Ca^2+^ NPs and CS-SA NPs, the microstructure, and the antioxidant and antibacterial properties of PE-coated D-limonene were also investigated. Furthermore, the antioxidant and antibacterial activity of free D-limonene and D-limonene PEs were further compared after heating or UV irradiation. This study provided a helpful reference for the application of CS-SA NPs-stabilized PE and improved the stability of D-limonene essential oil, which was conducive to expanding the application of D-limonene in the food industry.

## 2. Materials and Methods

### 2.1. Materials

The D-limonene (95%) was provided by Shanghai Aladdin Biochemical Technology Co., Ltd. (Shanghai, China). Low viscosity (100–200 mPa.s), medium viscosity (200–400 mPa.s), and high viscosity (>400 mPa.s) of CS, SA, CaCl_2_, and acetic acid were purchased from Shanghai Macklin Biochemical Technology Co., Ltd. (Shanghai, China). Medium-chain triglyceride (MCT) was supplied by Shanghai Yuanye Biotechnology Co., Ltd. (Shanghai, China). All reagent chemicals were analytical grade.

### 2.2. Preparation of CS-SA/Ca^2+^ NPs

CS-SA/Ca^2+^ NPs were prepared using the previously described method with minor modifications [26,27]. Briefly, the CS powder was dissolved in an acetic acid solution (1.0%, *v*/*v*), and the pH was adjusted to 5.4 using a 2 mol/L NaOH solution, which was recorded as solution A. The SA powder was dispersed in 40 mL of distilled water to obtain SA solution, and the pH was adjusted to 5.2 using a 0.1 mol/L HCl solution. Then, the SA solution was mixed with an 8 mL CaCl_2_ solution to form a pre-gel, which was recorded as solution B. Subsequently, 12 mL of solution A was added dropwise into solution B under magnetic stirring at 800 r/min for 1 h. Finally, the mixtures were dispersed with an ultrasonic homogenizer (Scientz-IID, Ningbo, China) at 262 W for 8 min in an ice bath. The nanoparticle suspension was freeze-dried after dialysis and placed in a silica gel dryer for usage. The effects of CaCl_2_ concentration (0–1.6 mg/mL), SA concentration (0.2–1.8 mg/mL), and CS concentration (0.2–1 mg/mL) on particle size, PDI, and zeta potential of NPs were investigated.

### 2.3. Characterization of CS-SA/Ca^2+^ NPs

#### 2.3.1. Particle Size, PDI and Zeta Potential

The average particle size, PDI, and zeta potential of NPs were measured using a particle size analyzer (Zetasizer Nano ZSE, Malvern, UK). Prior to measurement, the NPs suspension was diluted twice to avoid multiple scattering and meet the instrument sensitivity [28].

#### 2.3.2. Scanning Electron Microscopy (SEM)

The microstructure of NPs was imaged using SEM (Hitachi SU5000, Tokyo, Japan) according to a previously reported method with small modifications [29]. In summary, the NP freeze-dried powders were applied to the optical microscope stage, completely air-dried, and then treated with gold spray. The samples were observed at 15 kV to obtain the morphology of the NPs.

### 2.4. Preparation of Pickering Emulsions

The PEs of D-limonene were prepared by mixing CS-SA/Ca^2+^ NPs, D-limonene and MCT. Based on the team’s previous findings, 700 μL D-limonene and 300 μL MCT were added to the 10 mL NP suspension dropwise and stirred for 5 min at 800 r/min in an ice bath. Then, the mixture was homogenized by a homogenizer (HUXI, HR-25D, Shanghai, China) for 3 min at different rotational speeds. Finally, the emulsion was prepared with an ultrasonic homogenizer at 262 W for 5 min in an ice bath. The effects of NPs concentration (1.0–5.0 mg/mL) and homogenization speed (5000–25,000 r/min) on droplet size, and the zeta potential of D-limonene PEs were investigated.

### 2.5. Characterization of Pickering Emulsions

#### 2.5.1. Droplet Size and Zeta Potential

The Particle Size Analyzer (Zetasizer Nano ZSE, Malvern, UK) was used to measure the average droplet size and zeta potential. PEs were diluted with ultrapure water (1:100, *v*/*v*) to prevent multiple scattering.

#### 2.5.2. Creaming Index

The stability of D-limonene PEs was evaluated by the creaming index (*CI*). After centrifuging 9 mL of PE at 4 °C and 5000 r/min for 10 min, the emulsion was stratified. The formula for calculating *CI* was as follows:(1)$$CI\;\left(\%\right)=\frac{H_{s}}{H_{t}}\times 100$$
where *H_s_* and *H_t_* was the height of serum layer and total height, respectively.

#### 2.5.3. Optical Microscope

A drop of PE was pipetted and gently deposited onto a slide, and then photographed using an optical microscope B60F (Daoyi, Guangzhou, China) with a 40× objective lens.

#### 2.5.4. Confocal Laser Scanning Microscopy (CLSM)

The adsorption of CS-SA/Ca^2+^ NPs at the oil-water interface was observed by CLSM (IXplore spinSR, Tokyo, Japan). Specifically, 25 μL Nile blue (1%, *w*/*v*) and 20 μL Nile red (0.1%, *w*/*v*) were added to 1 mL of PE (diluted 4 times) to stain the NPs and oil phases, respectively, then kept in a dark place for 30 min. The excitation wavelengths of Nile red and Nile blue were 488 nm and 633 nm, respectively [30].

#### 2.5.5. Fourier-Transform Infrared (FTIR) Spectroscopy

The infrared spectra of CS, SA, CS-SA/Ca^2+^ NPs, CS-SA NPs, D-limonene, and PE were recorded using a FT-IR spectrometer (Tensor 27, Bruker Instrument, Karlsruhe, Germany) over the wavenumber range of 400 to 500 cm^−1^ [31]. The samples were mixed with dried KBr powder, ground, pressed, and then measured.

### 2.6. Antioxidant Activities

#### 2.6.1. 1,1-Diphenyl l-2-Pyridine Hydrazine (DPPH) Scanning Activity

The removal capacity of DPPH in the emulsions was according to the method of Lee et al. [32], with a slight modification. Samples (2 mL) were mixed with a 2 mL 0.1 mmol/L DPPH solution and left to stand for 30 min at room temperature in the dark. The absorbance was measured at 517 nm using the spectrophotometer (Varioskan LUX, Thermo Fisher Scientific, Waltham, MA, USA) after centrifugation. The test D-limonene PEs were prepared by diluting the original emulsion at different multiples. The DPPH scavenging activity (%) of the samples was calculated using Equation (2).
(2)$$Scavenging\;activity\left(\%\right)=\left(1-\frac{A_{1}-A_{2}}{A_{0}}\right)\times 100$$
where *A*_0_, *A*_1_, and *A*_2_ were the absorbances of the blank, the samples, and the control without DPPH, respectively.

#### 2.6.2. 2,2′-Azinobis-(3-Ethylbenzothiazoline-6-Sulfonic Acid) (ABTS) Radical Cation Scavenging Activity

The D-limonene PEs were mixed with an equal volume of ABTS solution, and the absorbance was measured at 713 nm after a 6 min reaction at room temperature [33]. The test D-limonene PEs were prepared by diluting the original emulsion at different multiples. The ABTS scavenging activity (%) was calculated using Equation (3).
(3)$$Scavenging\;activity\left(\%\right)=\left(1-\frac{A_{4}-A_{5}}{A_{3}}\right)\times 100$$
where *A*_3_, *A*_4_, and *A*_5_ were the absorbances of the blank, the test samples, and the control without ABTS, respectively.

### 2.7. Antibacterial Activity of Pickering Emulsions

#### 2.7.1. Antibacterial Activity Assay

The antibacterial activity of D-limonene PE was evaluated using the Oxford cup method [34]. Specifically, 100 μL bacterial suspension was added to 20 mL sterilized nutrient agar medium (*E. coli* and *S. aureus*) and solid Luria–Bertani (LB) medium (*S. putrefaciens*), mixed and poured into the 9 cm Petri dish. After solidifying, Oxford cups were removed, and 100 μL of samples (CS-SA/Ca^2+^ NPs suspension, free D-limonene and PEs) were added to the well. A total of 100 μL of sterile water was used as a blank group. In addition, the amount of free D-limonene added was consistent with the content of D-limonene in the PE. Finally, these bacteria were cultured in an incubator (THZ-702A, JD, Changzhou, China) at an appropriate temperature for 24 h (*E. coli* and *S. aureus* was cultured at 37 °C, and *S. putrefaciens* were cultured at 27 °C), and then the diameter of the antibacterial zones was measured. Finally, after sealing the Petri dish with a sealing film, it was put into a sealed bag and cultured in an incubator.

#### 2.7.2. SEM Observation of Bacteria

The effect of PE on the morphology of bacteria was observed by SEM. The experimental procedures were according to Qi et al.’s method [35], with some modifications. In short, *S. aureus*, *E. coli*, and *S. putrefaciens* at a concentration of 1 × 10^8^ CFU/mL were mixed with PE and incubated at 37 °C, 37 °C, and 27 °C for 8 h, respectively. Subsequently, all bacterial suspensions were centrifuged at 8000× *g* for 10 min at 4 °C (JIDI-20R, JIDI Instrument, Guangzhou, China). After cleaning the precipitates with phosphate-buffered saline (PBS, 0.1 M, pH 7.2), they were fixed in the glutaraldehyde fixing solution for 12 h. The bacteria were dehydrated with 20%, 40%, 60%, 80%, and 100% ethanol in turn. Finally, the samples were put on a silicon plate, sprayed with gold and then observed using SEM (Hitachi SU5000, Tokyo, Japan).

### 2.8. Effects of Heating and UV Light on the Antioxidant and Antibacterial Abilities of the PEs

10 mL of the freshly prepared emulsions and the same volume mixture of D-limonene, MCT and CANP suspensions were respectively put into screw-capped brown bottles with a diameter of 1.3 cm and placed in a water bath at 90 °C for 6 h in darkness, then the antioxidant and antibacterial activities were measured. In addition, the stability of D-limonene PE and free D-limonene under ultraviolet irradiation was evaluated. Specifically, two identical samples were added into transparent glass bottles with a diameter of 1.3 cm and irradiated under UV light at 395 nm (188 mW/cm^2^) in a camera obscura for 8 h to determine the changes in their antioxidant and antibacterial activities [36]. In order to eliminate the influence of water on D-limonene during heating and UV irradiation, the same volume of NP suspension as in the PEs was added to the free D-limonene, and then the treated free D-limonene was prepared into the emulsion. All samples were exposed to air and aqueous solutions.

### 2.9. Statistical Analysis

All experiments were performed at least three times, and the data were analyzed using SPSS software (SPSS 25.0, SPSS Inc., Chicago, IL, USA). The results were expressed in the form of the mean ± standard deviation (SD). One-way analysis of variance (ANOVA) and Duncan’s multiple range test were used to analyze the differences, which were considered to be significant at *p* < 0.05.

## 3. Results and Discussion

### 3.1. Characterization of CS-SA/Ca^2+^ NPs

#### 3.1.1. Particle size, PDI, and Zeta Potential of Nanoparticles

The effects of Ca^2+^ on nanoparticles are shown in Figure 1A,D,G. During the formation of CS-SA/Ca^2+^ NPs, Ca^2+^ acted as a gelation agent to promote SA to form Ca-alginate gel beads [37]. When Ca^2+^ was absent, the particle size and zeta potential of CS-SA NPs were 335 ± 6.88 nm and −44 ± 0.12 mV, respectively. Moreover, the particle size of CS-SA/Ca^2+^ NPs first decreased and then increased with the increase of Ca^2+^. When the concentration of CaCl_2_ was 0.2 mg/mL, the particle size was the smallest (268 ± 3.62 nm). The potential was gradually increased with an increase in CaCl_2_ concentration. When the CaCl_2_ concentration was 1.6 mg/mL, the potential reached −29 ± 0.37 mV. This could be attributed to the fact that Ca^2+^ not only binds with guluronic acid fragments of SA but can also be chelated by SA. When the content of Ca^2+^ was low, a large amount of CS and SA were complexed together due to electrostatic interaction, resulting in a larger particle size, while excess Ca^2+^ induced more CS and SA to react with it, resulting in an increase in particle size and zeta potential.

Moreover, the stability of the CS-SA/Ca^2+^ NPs also depends on the concentration of SA and CS. It is worth noting that large particles were formed regardless of whether the concentration of CS and SA was too high or too low. The low concentrations of CS and SA provided a small quantity of molecules available for complexation, leading to a weak interaction and resulting in larger and more incomplete CS-SA/Ca^2+^ NPs. Similarly, high concentrations of polymers encouraged the building of large aggregates. When the concentration of SA was 0.6 mg/mL and CS was 0.4 mg/mL, the minimum particle size of 274 ± 2.51 nm nanoparticles could be obtained. More importantly, it was also found from Figure 1B,C that too high or too low polymer concentration would lead to multiple peaks in the size distribution diagrams, indicating that the particle size of CS-SA/Ca^2+^ NPs at this concentration was inhomogeneity and the PDI was high (Figure 1E,F). As a hydrophilic anionic polymer, the zeta potential decreased from −29 ± 0.75 mV to −49 ± 0.69 mV with the increase of SA (Figure 1H). In contrast, the potential of the CS-SA/Ca^2+^ NP suspension decreased with the increase in CS concentration (Figure 1I). To sum up, the CS-SA/Ca^2+^ NPs prepared with 0.2 mg/mL CaCl_2_, 0.6 mg/mL SA, and 0.4 mg/mL CS had smaller particle size and PDI, as well as higher stability.

#### 3.1.2. Micromorphology of Nanoparticles

The microscopic morphology of CS-SA NPs and CS-SA/Ca^2+^ NPs was observed by SEM, and the results are shown in Figure 2A,B. As could be seen from the figure, CS-SA NPs presented irregular particle aggregates, and CS-SA/Ca^2+^ NPs were spherical particles, distinct and regular. Actually, Ca^2+^ cross-linked with the guluronic acid of SA to form a pre-gel of the “egg-box” model, and then the ionic interactions happened with CS to form spherical nanoparticles.

#### 3.1.3. Chemical Interactions of Nanoparticles

The chemical interactions among the components of CS-SA/Ca^2+^ NPs were investigated by FTIR (Figure 2C). In the infrared spectrogram of CS, some major peaks were observed at 3423 cm^−1^, 2876 cm^−1^, 1650 cm^−1^, and 1600 cm^−1^, which could correspond to the stretching of amine and hydroxyl groups (overlapped), −OH stretching, C=O stretching of the amide bond, and the bending vibration of the N-H (amide II band) at 1600 cm^−1^, respectively [38,39]. The stretching of N-H at 1422 and 1382 cm^−1^ (amide III band) and the characteristic peak of C-O stretching were observed at 1085 cm^−1^. In the SA spectrum, the absorption band at 3426 cm^−1^ corresponded to the stretching vibration of -OH groups. In addition, the typical bands at 1619 cm^−1^ and 1420 cm^−1^ were described as asymmetric and symmetric stretching of the carboxylate group, respectively. The absorption bands at 1028 cm^−1^ were the characteristic bands of C-O-C groups [22,40,41]. In the FTIR spectra of the CS and SA complexes, the absorption peaks of the amide group (1650 cm^−1^ and 1600 cm^−1^) and carboxylate group (1619 cm^−1^) disappeared due to the electrostatic interaction between the −NH^3+^ and −COO^-^ ions, also indicating that polyelectrolyte complexes were successfully formed [42]. By comparing the spectrum of SA, the stretching vibration peak of OH^-^ moved to 3453 cm^−1^ and the peak was sharper and more intense, as well as the stretching vibration peak of COO^−^ moved to 1702 cm^−1^–1643 cm^−1^ in the higher band, which was due to the cross-linking between CaCl_2_ and the GG segment of SA, forming an “eggshell” structure. This network structure changed the amplitude of COO^−^ and OH^−^ stretching vibrations on the macromolecular six-member ring, resulting in a shift in peak position [43].

### 3.2. Characterization of D-Limonene PEs

#### 3.2.1. Droplet Size, Zeta Potential, CI, and Optical Microscope Images of D-Limonene PEs

The effects of CS-SA/Ca^2+^ NP concentration and homogenization speed on the average droplet size, and zeta potential of PEs are shown in Figure 3. The results indicated that the droplet size decreased with the initial increase of CS-SA/Ca^2+^ NP content. This could be explained by the fact that the higher the concentration in CS-SA/Ca^2+^ NP within a certain range, the more nanoparticles are distributed at the oil/water interface, thus forming a compact interfacial film that hinders the coalescence between the droplets [44]. However, the excessive CS-SA/Ca^2+^ NPs formed multi-layer adsorption on the surface of the oil droplet, which increased the volume of the droplet. In particular, when the CS-SA/Ca^2+^ NP concentration reached 2 mg/mL, the droplet size was the smallest (521.88 ± 5.48 nm). In addition, it should be noted that the average D-limonene PE droplet size decreased from 606 ± 2.64 nm to 535 ± 16.39 nm as homogenization speed increased from 5000 r/min to 25,000 r/min at a constant nanoparticle concentration. It was known that the oil particle size of the D-limonene PE decreased with the increase in homogenization speed, while the total interface area of droplets increased accordingly, so more CS-SA/Ca^2+^ NPs needed to be adsorbed at the oil-water interface to prevent their aggregation [30]. When the homogenization speed reached 20,000 r/min, the droplet size was 550 ± 8.26 nm. However, the particle size remained unchanged as the homogenization speed increased again, and it was not hard to imagine that the amount of nanoparticles at this time was not enough to completely adsorb at the droplet interface, which caused the oil droplets to aggregate. Secondly, it should be noted that when the homogenization speed increased from 5000 r/min to 10,000 r/min, the droplet size increased to 680 ± 5.79 nm, which can be attributed to the fact that the homogenization speed of 5000 r/min did not fully emulsify the oil phase, resulting in an overall droplet size that was small and fairly uneven.

Figure 3B,E show the zeta potential of D-limonene PEs. Due to the negative charge characteristics of CS-SA/Ca^2+^ NPs, the more nanoparticles adsorbed on the surface of the oil droplet, the smaller the potential. When the absolute value of zeta potential was greater than or close to 30 mV, stronger electrostatic repulsion reduced the aggregation of droplets and ensured the stability of the system [22]. Therefore, a 2 mg/mL CS-SA/Ca^2+^ NP concentration and a 20,000 r/min homogenization rate were considered as the formulation of D-limonene PE.

Figure 3G,H represent optical microscope images of D-limonene PEs at different CS-SA/Ca^2+^ NP concentrations and homogenization speeds, respectively. As can be seen from Figure 3G,H, the CS-SA/Ca^2+^ NPs adsorbed on the surface of the oil droplets, preventing the aggregation between the oil droplet and the oil droplet and giving it a spherical structure. This phenomenon showed that CS-SA/Ca^2+^ NPs could be used as a good emulsifier to stabilize D-limonene PE.

#### 3.2.2. Creaming Index

Due to the density difference between D-limonene and the aqueous phase, the emulsion was stratified; that is, the upper layer was emulsion and the lower layer was water. The creaming index (CI) was often used to characterize the stability of emulsions, and the creaming indexes of D-limonene PE were shown in Figure 3C,F. The stability of PE was improved with the increase in CS-SA/Ca^2+^ NP concentration. When the NP concentration was 2 mg/mL, the CI value decreased by 10.33%, indicating that the CS-SA/Ca^2+^ NPs hindered the collision and coalescence of oil droplets and controlled the phase separation of the D-limonene PE. However, with the continuous increase of NPs, the droplet size increased, and the emulsion was affected by sedimentation, which led to the increase of CI. In addition, at the homogeneous speed of 20,000 r/min, the CI was 27.9 ± 3.94%, which was also considered to be related to the smaller particle size.

#### 3.2.3. Microscopic Morphologies of the D-Limonene PEs

The interfacial structure of PEs was further analyzed using CLSM. CS-SA/Ca^2+^ NPs are shown in red (Figure 4A), and the oil phase was represented in green (Figure 4B), and Figure 4C exhibits the overlapping images. It could be observed that the CS-SA/Ca^2+^ NPs adsorbed on the surface of the oil droplets formed an interface film, indicating that the D-limonene was W/O type [45]. This also showed that the CS-SA/Ca^2+^ NPs changed the hydrophobicity of D-limonene essential oil, providing more possibilities for its application in food.

#### 3.2.4. FTIR Analysis of PEs

FTIR analysis was employed to evaluate the interactions between CS-SA/Ca^2+^ NP and D-limonene. The FTIR spectra of CS-SA/Ca^2+^ NP, D-limonene, PE, and MCT were shown in Figure 4D. For D-limonene, the peaks related to the stretching vibration of C-H were found at 2919 cm^−1^ and 1439 cm^−1^, as well as the characteristic peak of C=C stretching presented at 1643 cm^−1^. Moreover, the two obvious peaks at 886 cm^−1^ and 795 cm^−1^ showed the bending vibration of =C–H [46]. In the FTIR spectra of PE, it could be seen that several characteristic peaks of D-limonene (1339, 886, and 795 cm^−1^) disappeared almost completely, indicating that D-limonene was successfully wrapped in the CANP droplets [47]. Similarly, Kou et al. [48] verified the phenomenon that the stretching bonds of C-C and C-H on the benzene ring peaks of ginger oil disappeared after being wrapped in β-cyclodextrin. Overall, there were no new chemical bond transmission peaks in the PE spectrum. Therefore, it could be speculated that there was only a simple physical mixing between the CS-SA/Ca^2+^ NP and D-limonene and no new chemical bonds emerged.

#### 3.2.5. Antioxidant Activities of D-Limonene PEs

The antioxidant activity of D-Limonene PE was evaluated using DPPH and ABTS radical scavenging ratio tests. The current studies suggest that D-limonene essential oil possesses certain antioxidant capacity [49]. The aim of this experiment was to discuss the change in antioxidant activity after D-limonene was prepared into PE. Figure 5A confirmed that the D-limonene PE had excellent scavenging ability against DPPH and ABTS free radicals, and its scavenging ability was comparable to that of a 5 mg/mL vitamin C solution. After diluting D-limonene PE by 8 times (20%), it still had 62 ± 5.89% (DPPH) and 60 ± 1.16% (ABTS) scavenging capacity. In addition, a free D-limonene-ethanol solution with the same concentration as 20% D-limonene PE was prepared, and the free radical scavenging experiment was carried out. The results showed that it could remove 24 ± 1.26% of DPPH and 37 ± 1.48% of ABTS free radicals. In contrast, D-limonene PE had a higher clearance activity than free D-limonene. This may be understood as emulsification improving the hydrophobicity of D-limonene essential oil, enhancing dispersibility, which can promote full contact between D-limonene and free radicals, and helping to improve its ability to capture free radicals [50].

During the storage process, a series of chain reactions of free radicals can be induced due to the free radicals produced in essential oil under conditions of heat or ultraviolet (UV) light exposure, resulting in severe oxidation of limonene [36]. Figure 5B studied the protective effect of CS-SA/Ca^2+^ NPs on the antioxidant activity of D-limonene essential oil under high temperature and UV conditions. According to the figure, although the antioxidant capacity of D-limonene PE was reduced after high temperatures and UV treatment, it still had better antioxidant activity than free D-limonene. The DPPH and ABTS clearance rates of D-limonene PE after heating were 87 ± 0.64% and 83 ± 0.15%, respectively, while the DPPH and ABTS clearance rates of free D-limonene after heating were 70 ± 6.66% and 77 ± 1.09%, respectively. On the other hand, the DPPH and ABTS clearance rates of D-limonene PE after UV irradiation were 83 ± 0.49% and 98 ± 0.91%, while the DPPH and ABTS clearance rates of limonene emulsion after heating were 44 ± 3.20% and 95 ± 0.42%, respectively. The results showed that D-limonene was more sensitive to temperature than UV irradiation and also indicated that PE had a good protective effect on free D-limonene.

#### 3.2.6. Antibacterials Abilities of D-Limonene PEs

The inhibition zone test was performed to study the antibacterial action of D-limonene PE. *S. aureus* was selected as a model of Gram-positive bacteria, and *E. coli* and *S. putreficans* as Gram-negative bacteria. The original D-limonene PE was chosen as the test sample, the CS-SA/Ca^2+^ NP suspension and free D-limonene were used as the controls, and water was used as the blank control. The results showed that *E. coli*, *S. putreficans*, and *S. aureus* all observed an obvious inhibition zone after contact with PE. According to the diameter of the inhibition zone, the sensitivity of the tested bacteria to D-limonene PE was *S. putrefaciens* (24.96 ± 0.25 mm) > *E. coli* (18.84 ± 0.20 mm) > *S. aureus* (15.10 ± 0.70 mm), which indicated that D-limonene PE could be used as a good bacteriostatic agent. More importantly, by comparing the antibacterial properties of PE and free D-limonene, it was also found that the antibacterial effects of PE on the three tested bacteria were stronger than those of free D-limonene. Compared the bacteria inhibition zone of free D-limonene with that of PE-protected D-limonene, the inhibition zone increased from 18.84 ± 0.20 to 24.96 ± 0.25 mm (*E. coli*), 24.05 ± 0.78 to 26.55 ± 0.17 mm (*S. putrefaciens*), and 11.58 ± 0.76 to 15.10 ± 0.70 mm (*S. aureus*), increased by 32.48%, 10.40%, and 30.40%, respectively. This consequence might be related to the following reasons: Firstly, the CS-SA/Ca^2+^ NPs emulsified D-limonene and endowed PE with superior hydrophilicity, which was conducive to the infiltration of D-limonene into the biofilm of bacteria. Secondly, the prepared PE had a smaller droplet size, making it easier to contact the cell membranes of bacteria. Thirdly, CS-SA/Ca^2+^ NPs might have a certain slow-release and protective effect on D-limonene essential oil, thereby reducing the release speed of essential oil and extending its bactericidal performance [34,51].

In order to further investigate the protective effect of CS-SA/Ca^2+^ NPs on the antibacterial properties of D-limonene, PE and free D-limonene were treated by heating and ultraviolet irradiation to evaluate their activities. Similarly, as could be seen from Figure 6B, after heating treatment, the inhibitory zones of free D-limonene against *E. coli*, *S. putrefaciens* and *S. aureus* were 13.06 ± 0.42 mm, 20.86 ± 1.3 mm, and 9.29 ± 0.20 mm, respectively, while the inhibitory zones of the D-limonene emulsion were 18.76 ± 0.13, 24.17 ± 1.14, and 12.18 ± 0.41 mm. This showed that the CS-SA/Ca^2+^ NPs could effectively reduce the effect of high temperature on the bacteriostatic effect of D-limonene. Figure 6B also showed that the effect of ultraviolet irradiation on the antibacterial properties of D-limonene was less than that of heating treatment. There was no significant difference in the inhibition effect of free D-limonene and PE on *S. putrefaciens* and *S. aureus* after ultraviolet irradiation, but the inhibition effect of PE (24.68 ± 0.15 mm) on *E. coli* was still better than that of free D-limonene (21.98 ± 1.25 mm). The protective effect of the PE on D-limonene was assumed to be that the CS-SA/Ca^2+^ NPs formed a three-dimensional mesh interface barrier on the droplet surface, and the oil droplets were surrounded by a dense multi-layer of CANPs, which lessened the direct entry of light into the oil droplet surface, thus effectively protecting D-limonene against UV light [52,53].

#### 3.2.7. SEM Observation of Bacteria

To analyze the antibacterial mechanism of D-limonene PE, the morphological changes of three types of bacteria before and after treatment were observed by SEM. The optical morphology of *E. coli* showed a short rod-like structure, *S. putrefaciens* was long rod-like, and *S. aureus* was spherical [54]. In the control group, they had smooth surfaces and intact structures. However, under the action of D-limonene PE, the structure of these bacteria showed a certain degree of damage. Among them, *E. coli* and *S. putrefaciens* displayed an obvious morphological alteration. Specifically, the cell wall was damaged, which led to the release of cytoplasmic inclusions; serious aggregation and adhesion occurred, resulting in the death of the bacteria. Furthermore, some *S. aureus* showed distinct folds and depressions in their cell membranes. This result suggested that the inhibition effects of the D-limonene PE on the three types of bacteria were probably due to the fact that the emulsion damaged their cytoplasmic membranes and increased membrane permeability [55].

## 4. Conclusions

The majority of studies on PEs stabilized by chitosan-based nanoparticles involved the complexation of CS with proteins; only a few studies reported the production of PEs stabilized by complex nanoparticles formed by CS and other polysaccharides, especially the application of CS-SA/Ca^2+^ NPs stabilized PE for the encapsulation, protection, and delivery of essential oils. In this study, CS-SA/Ca^2+^ NPs were used as a promising emulsifying stabilizer to prepare PEs with good thermal and ultraviolet protection effects on D-limonene. The spherical structure of the CS-SA/Ca^2+^ NPs benefited from the chemical cross-linking between Ca^2+^ and SA. Moreover, due to the introduction of CS-SA/Ca^2+^ NPs, the interfacial surface and network structure formed by nanoparticles acted as a barrier at the oil-water interface, reducing the direct entry of light and weakening the effect of heating on the activity of D-limonene essential oil. In addition, the water solubility of the encased D-limonene was greatly improved. The substant PE had a good protection effect on D-limonene under heat and UV irradiation. The antioxidant property was comparable to that of vitamin C at the PE concentration of %, and the antimicrobial effects on bacteria were *S. putrefaciens > E. coli > S. aureus*, and the antimicrobial activities were increased compared with free D-limonene. In addition, the PE protected D-limonene well against heat and UV. Compared to other types of PEs, CS-SA NP-stabilized D-limonene emulsion had a smaller droplet size and was more conducive to the active substance penetrating the barrier and acting. Therefore, the surfactant-free nature of the PE stabilized by CS-SA/Ca^2+^ NPs further reflects its potential as a safe delivery system that can effectively envelop lipophilic substances and protect their biological activity, with potential food applications.

## Figures and Tables

**Figure 1 foods-13-00622-f001:**
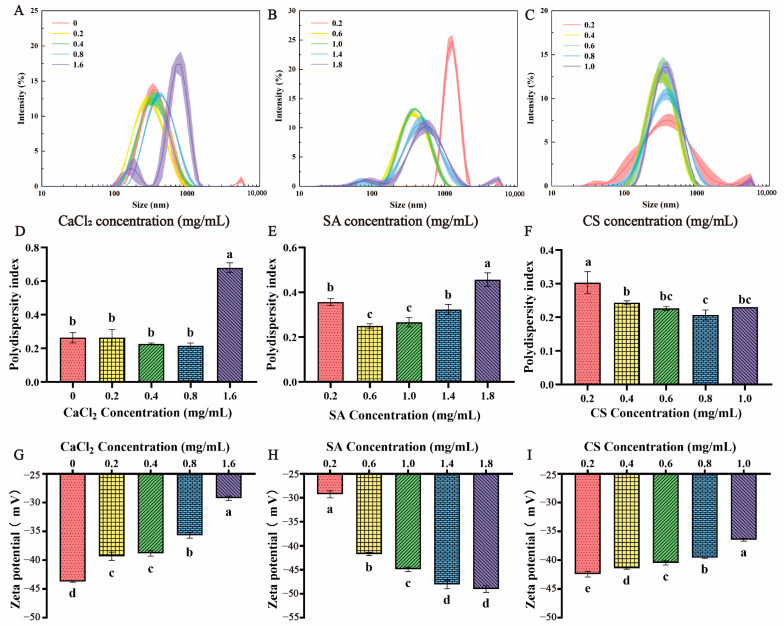
(**A**,**D**,**G**) represented the particle size, PDI and zeta potential of nanoparticles at 0.2–1.0 mg/mL CaCl_2_, 0.6 mg/mL SA, and 0.4 mg/mL CS; Figure (**B**,**E**,**H**) represented the particle size, PDI and zeta potential of nanoparticles at 0.2–1.8 mg/mL SA, 0.2 mg/mL CaCl_2_, and 0.4 mg/mL CS; Figure (**C**,**F**,**I**) represented the particle size, PDI and zeta potential of nanoparticles at 0.2–1.0 mg/mL CS, 0.2 mg/mL CaCl_2_, and 0.4 mg/mL CS. Values with different superscripts indicate significant differences (*p* < 0.05).

**Figure 2 foods-13-00622-f002:**
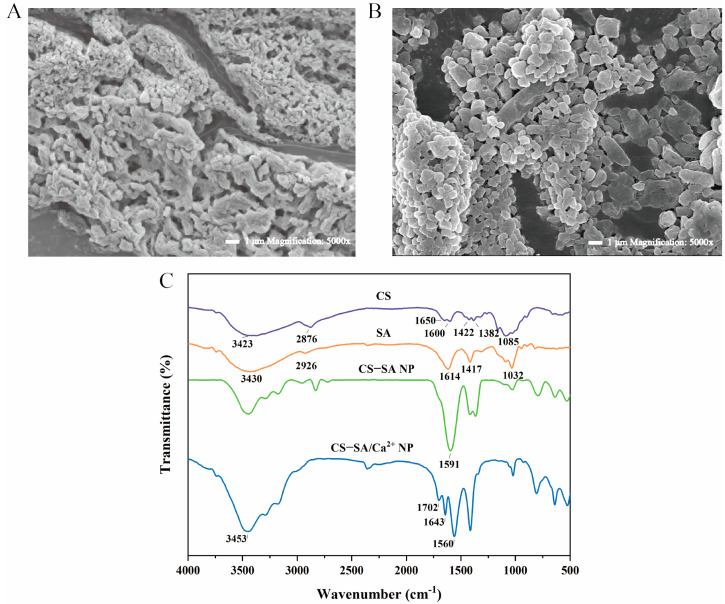
(**A**) Micromorphology of chitosan-alginate nanoparticles (CS-SA NPs) at 5000×; (**B**) Micromorphology of chitosan-alginate/CaCl_2_ nanoparticles (CS-SA/Ca^2+^ NPs) at 5000×; (**C**) Fourier-transform infrared (FTIR) spectra of CS, SA, CS-SA NP, and CS-SA/Ca^2+^ NPs.

**Figure 3 foods-13-00622-f003:**
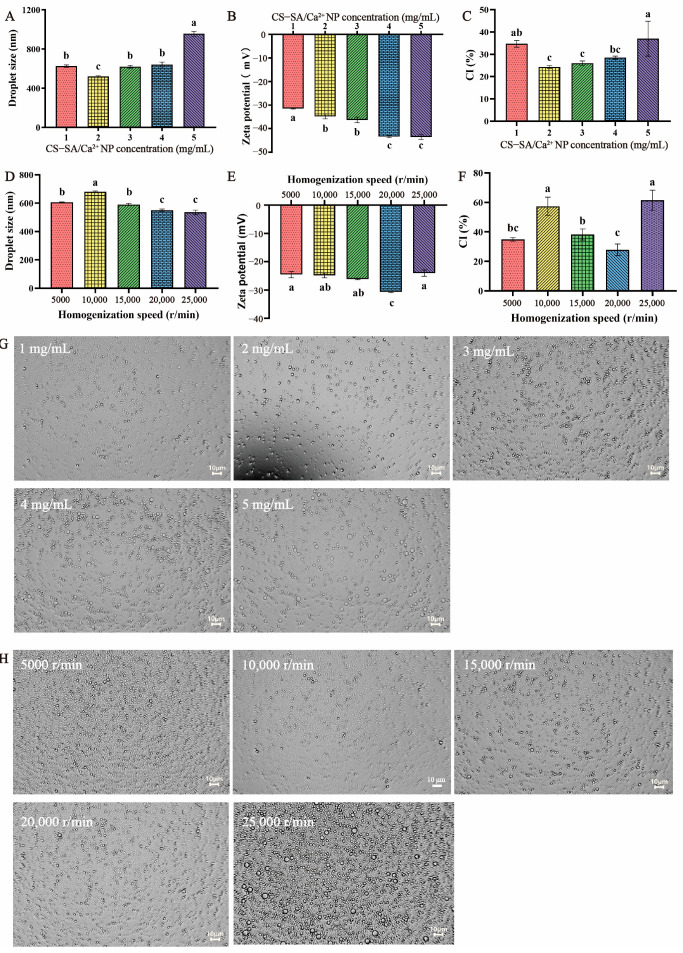
(**A**–**C**) Droplet size, zeta potential, and creaming index (CI) of D-limonene Pickering emulsions (PEs) at different CS-SA/Ca^2+^ nanoparticle concentrations; (**D**–**F**) Droplet size, zeta potential, and creaming index (CI) of D-limonene Pickering emulsions (PEs) at different homogenization speed; (**G**) Optical microscope images of D-limonene Pickering emulsions (PEs) with CS-SA/Ca^2+^ nanoparticle concentration of 1 mg/mL, 2 mg/mL, 3 mg/mL, 4 mg/mL, 5 mg/mL; (**H**) Optical microscope images of D-limonene Pickering emulsions (PEs) with homogenization speed of 5000 r/min, 10,000 r/min, 15,000 r/min, 20,000 r/min, 25,000 r/min. Values with different superscripts indicate significant differences (*p* < 0.05).

**Figure 4 foods-13-00622-f004:**
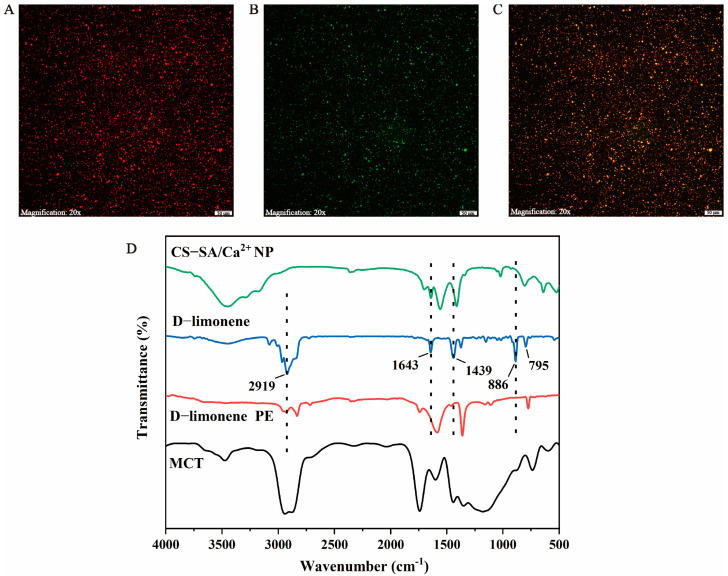
(**A**–**C**) The confocal laser scanning microscopy (CLSM) images of the D-limonene PE; (**D**) Fourier-transform infrared (FTIR) spectra of chitosan/alginate nanoparticle (CS-SA/Ca^2+^ NP), D-limonene, D-limonene Pickering emulsion (PE), and medium chain triglyceride (MCT).

**Figure 5 foods-13-00622-f005:**
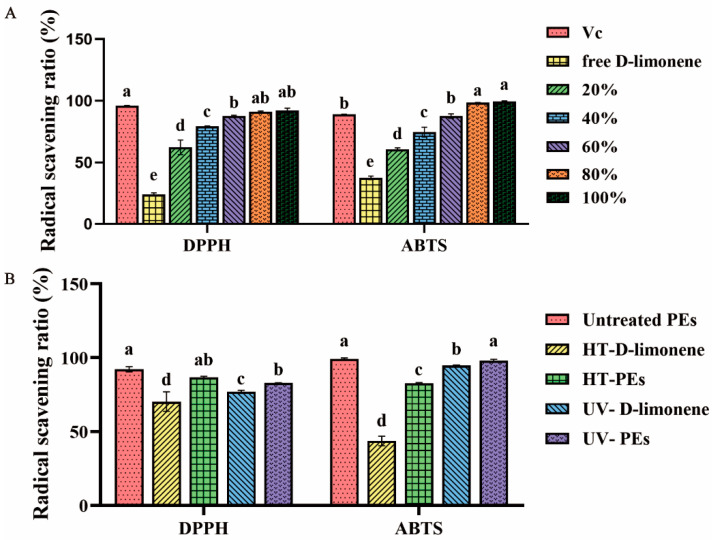
(**A**) DPPH and ABTS radical scavenging activities of 5 mg/mL vitamin C (Vc), free D-limonene, 20%, 40%, 60%, 80%, and 100% D-limonene PE; (**B**) DPPH and ABTS clearance rate of untreated PE, high temperature treated D-limonene (HT-D-limonene), high temperature treated PE (HT-PE), UV irradiation D-limonene (UV-D-limonene) and UV. Values with different superscripts indicate significant differences (*p* < 0.05).

**Figure 6 foods-13-00622-f006:**
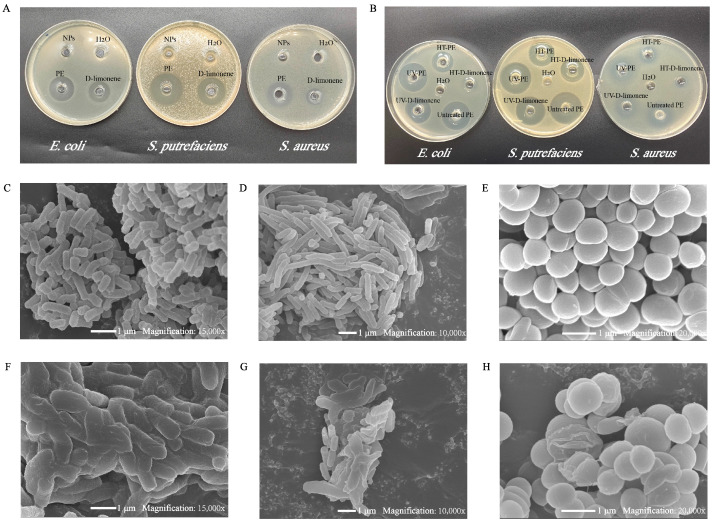
(**A**) Antibacterial activities of H_2_O, D-limonene, CS-SA/Ca^2+^ NPs, and D-limonene PE; (**B**) Antibacterial activities of untreated PE, HT-D-limonene, HT-PE, UV-D-limonene, and UV-PE; scanning electronic microscope (SEM) images of *E. coli* (**C**,**F**), *S. putrefaciens* (**D**,**G**), and *S. aureus* (**E**,**H**) before and after the D-limonene Pickering emulsion treatment.

## Data Availability

Data are contained within the article, further inquiries can be directed to the corresponding author (Zhong, S.).

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
