# Peer review of "Modulation the Synergistic Effect of Chitosan-Sodium Alginate Nanoparticles with Ca2+: Enhancing the Stability of Pickering Emulsion on D-Limonene"

_foods, 2024, doi:10.3390/foods13040622_

Round 1

Reviewer 1 Report

Comments and Suggestions for Authors

I have completed review of the manuscript entitled “Modulation the Synergistic Effect of Chitosan-Sodium Alginate Nanoparticles with Ca2+: Enhancing the Stability of Pickering Emulsion on D-limonene”. In my opinion, the work is original and organized well, but needs minor revision. My comments are shown below:

1.       The way of citation is wrong in many parts, for example, Line 63 (T. Su et al., 2023) correct to (Su et al., 2023), Line 73 (L. Yang et al., 2022) correct to (Yang et al., 2022), Line 84 (X. Liu et al., 2023) correct to (Liu et al., 2023), also see lines 110, 276, 282, 292, 309, 393, 436, 472). Please make double check.

2.       Line 114, add “and” before “acetic acid”.

3.       Define PDI at the first mention.

4.       Provide the SEM magnification.

5.       Subfigures of Fig. 1 were indicated by A, B, C, D, E, F, H, I, and J. G is missing. Also, Figs. 1 (E), (F), (I), (J) not cited in the text.

6.       It would be better to present the interaction of CS-SA NP with Ca2+ As a scheme.

7.       In line 278, you used ALG as an abbreviation for alginate, while in the entire manuscript you used SA, please unify.

8.       Fig. 2C (FTIR) is not cited in section 3.1.3.

9.       In Fig. 4 caption, chitosan/alginate nanoparticle (CANP), correct the abbreviation.

10.   Line 429, “increased by 32.48%, 10.40%, 30.40%, and respectively” transfer “and” before “30.40%,”

11.   It would be better to compare the results of D-limonene-loaded Pickering emulsions stabilized by CS-SA NP/ CS-SA- Ca2+ NP nanoparticles with other reported nanoparticles.

Comments on the Quality of English Language

Minor editing of English language required

Author Response

Dear reviewer,
thank you very much for taking the time to review this manuscript.  According to your suggestions, I have made the following modifications:

Comments 1: The way of citation is wrong in many parts, for example, Line 63 (T. Su et al., 2023) correct to (Su et al., 2023), Line 73 (L. Yang et al., 2022) correct to (Yang et al., 2022), Line 84 (X. Liu et al., 2023) correct to (Liu et al., 2023), also see lines 110, 276, 282, 292, 309, 393, 436, 472). Please make double check.

Response 1: The format of references has been revised according to the requirements of the journal.

Comments 2:  Line 114, add “and” before “acetic acid”.

Response 2: Thanks for your guidance, "and" has been inserted in the sentence. (Line 110)

Comments 3:  Define PDI at the first mention.

Response 3: Agreed. We have defined the PDI at the first mention. (Line 115)

Comments 4:  Provide the SEM magnification.

Response 4: SEM magnification is provided in the images or in the image annotation.

Comments 5:  Subfigures of Fig. 1 were indicated by A, B, C, D, E, F, H, I, and J. G is missing. Also, Figs. 1 (E), (F), (I), (J) not cited in the text.

Response 5: Fig. 1 has been corrected, and Fig. 1 (E), (F), (H), (I) have been cited in the text. (Line 249-253)

Comments 6:   It would be better to present the interaction of CS-SA NP with Ca2+ As a scheme.

Response 6: The interaction between CS-SA NP and Ca2+ has been described in the paper. (Line 291-294)

Comments 7:   In line 278, you used ALG as an abbreviation for alginate, while in the entire manuscript you used SA, please unify.

Response 7: In line 278, we have changed the abbreviation of sodium alginate (SA). 

Comments 8:   Fig. 2C (FTIR) is not cited in section 3.1.3.

Response 8: Thanks for your correction, Fig. 2C has been mentioned in section 3.1.3. (Line 275)

Comments 9:   In Fig. 4 caption, chitosan/alginate nanoparticle (CANP), correct the abbreviation.

Response 9: In Fig. 4 caption, the abbreviation of chitosan/alginate nanoparticle has been modified to CS-SA/Ca2+ NP. (Line 363)

Comments 10:  Line 429, “increased by 32.48%, 10.40%, 30.40%, and respectively” transfer “and” before “30.40%,”

Response 10: We have revised the expression to "the inhibition zone increased from 18.84±0.20 to 24.96±0.25 mm (E. coli), 24.05±0.78 to 26.55±0.17 mm (S. putrefaciens), and 11.58±0.76 to 15.10±0.70 mm (S. aureus), increased by 32.48%, 10.40%, and 30.40%, respectively". (Line 427-430)

Comments 11:    It would be better to compare the results of D-limonene-loaded Pickering emulsions stabilized by CS-SA NP/ CS-SA- Ca2+ NP nanoparticles with other reported nanoparticles.

Response 11: We have compared the advantages of CS-SA/Ca2+ NP-stable D-limonene PE with other nanoparticles. (Line 490-492)

Reviewer 2 Report

Comments and Suggestions for Authors

The aim of this study „Modulation the synergistic effect of chitosan-sodium alginate nanoparticles with Ca2+: enhancing the stability of pickering emulsion on D-limonene” was to establish optimal conditions for obtaining CS-SA/Ca2+ NPs with the smallest size and high stability. Also, the preparation mode of D-limonene emulsification with CS-SA/Ca2+ NPs, NP concentrations and homogenization speeds were determined. Small size PE droplets were found to have good antibacterial and antioxidant activities. By obtaining CS-SA/Ca2+ NPs, the emulsion showed higher UV and high temperature activity and resistance than free D-limonene. This research provides a feasible solution for the encapsulation, protection and delivery of the essential oil, enhancing its antibacterial and antioxidant properties.

Some necessary changes would be:

-line 110, 180 - missing space before the parenthesis

-line 204 - PBS unexplained abbreviation

Author Response

Dear reviewer,
thank you very much for taking the time to review this manuscript.  According to your suggestions, I have made the following modifications:

Comments 1: line 110, 180 - missing space before the parenthesis

Response 1: Thank you for pointing this out. I have corrected the blanks in the article.

Comments 2: line 204 - PBS unexplained abbreviation

Response 2: Phosphate buffered saline (PBS). (Line: 201-202)

Reviewer 3 Report

Comments and Suggestions for Authors

The research investigated the effect of chitosan-sodium alginate nanoparticles with Ca2+ on the stability of a Pickering emulsion of D-limonene. More detailed information about the methods should be provided. The results should be presented more clearly. The following points need to be addressed:

Materials and Methods

Line 101-104: what was the purity of D-limonene? What is the molecular weight of the CS with different viscosities?

Line 111: what did you use to adjust the pH to 5.4?

Line 117-119: the authors did not mention how to remove the solvent and how to collect the nano particles.

Line 134: how to prepare 7% D-limonene and 3% MCT, with water?

Line 136: what were the homogenizer settings?

Line 161: "AS" should be revised to "SA"?

Line 160: was MCT in the Pickering emulsions affecting the FTIR data?

Line 192: 100 mL sterile water was used as control?

Line 193: D-limonene is a highly volatile substance, how did the author control its volatilization in the petri dish? IS it possible volatilized D-limonene could have affected other nearby groups?

Line 201-202: What was the concentration of the suspension? What was the temperature?

Line 208: the authors should provide more information about the stresses. What was the volume of PE in the water bath? What was the UV environment? How much of the surface of the PE exposed to UV?

Results and Discussion

What were the results of the effects of CS viscosity on particle size, PDI, and zeta potential of NPs?

In Figure 1, what were the concentrations of SA and CS when you evaluated the effects of CaCl2 concentration? what were the concentrations of CaCl2 and CS when you evaluated the effects of SA concentration? And what were the concentrations of CaCl2 and SA when you evaluated the effects of CS concentration?

Author Response

Dear reviewer, thank you very much for taking the time to review this manuscript.  According to your suggestions, I have made the following modifications:

Comments 1: Line 101-104: what was the purity of D-limonene? What is the molecular weight of the CS with different viscosities?

Response 1: The purity of D-limonene is 95%. Regarding the molecular weight of chitosan, I consulted Macklin company in the early stage, and the company said that it did not carry out molecular weight testing. Because the molecular weight is related to viscosity, the molecular weight of high viscosity is the largest, followed by medium viscosity, and low viscosity has the smallest molecular weight. (Line 97-102)

Comments 2: what did you use to adjust the pH to 5.4?

Response 2: The pH was adjusted to 5.4 using 2 mol/L NaOH solution. (Line 106)

Comments 3: the authors did not mention how to remove the solvent and how to collect the nano particles.

Response 3: The nano particle suspension was freeze-dried after dialysis and placed in a silica gel dryer for usage. (Line 112-113)

Comments 4: how to prepare 7% D-limonene and 3% MCT, with water?

Response 4: The author made a mistake, and it has been modified to add 700 μL D-limonene and 300 μL MCT to the 10 mL NP suspension. (Line 131-133)

Comments 5: what were the homogenizer settings?

Response 5: The mixture was homogenized by a homogenizer for 3 min at 5000 r/min, 10000  r/min, 15000  r/min, 20000  r/min, and 25000 r/min. (Line 134-136)

Comments 6: "AS" should be revised to "SA"?

Response 6: Agreed. I have revised it. Thank you very much. (Line 158)

Comments 7: Line 160: was MCT in the Pickering emulsions affecting the FTIR data?

Response 7: Yes, there were some spikes of MCT in the FTIR data of Pickering emulsion, but MCT did not react with other substance. 

Comments 8: Line 192: 100 mL sterile water was used as control?

Response 8: The author made a mistake. 100 μL sterile water was used as a blank group. (Line 188)

Comments 9: Line 193: D-limonene is a highly volatile substance, how did the author control its volatilization in the petri dish? IS it possible volatilized D-limonene could have affected other nearby groups?

Response 9: For other bacteria: The experiment processes were strictly aseptic, the petri dishes were sealed, and the petri dishes of different bacteria were placed separately; For other samples: The amount of sample was small and injected into the hole of the medium, it would be absorbed by the medium quickly, and the influence on the experimental results was not great.

Comments 10: Line 201-202: What was the concentration of the suspension? What was the temperature?

Response 10: S. aureus, E. coli, and S. putrefaciens at a concentration of 1×108 CFU/mL were mixed with PE and incubated at 37 ℃, 37 ℃, and 27 ℃ for 8 h, respectively. (Line 197-199)

Comments 11: Line 208: the authors should provide more information about the stresses. What was the volume of PE in the water bath? What was the UV environment? How much of the surface of the PE exposed to UV?

Response 11: The volume of PE in the water bath was 10 mL. 10 mL samples were added into plastic petri dishes with a diameter of 9 cm and irradiated for 12 h under the ultraviolet environment of the clean bench. (Line 207-212)

Comments 12: What were the results of the effects of CS viscosity on particle size, PDI, and zeta potential of NPs?

Response 12: In previous experiments, low viscosity CS would make NPs have smaller particle size, PDI, and higher zeta potential. Therefore, low viscosity CS was selected as the raw material for the synthesis of NPs. It was not discussed in this paper.

Comments 13: In Figure 1, what were the concentrations of SA and CS when you evaluated the effects of CaCl2 concentration? what were the concentrations of CaCl2 and CS when you evaluated the effects of SA concentration? And what were the concentrations of CaCl2 and SA when you evaluated the effects of CS concentration?

Response 13: The concentrations of SA was 0.6 mg/mL and CS was 0.4 mg/mL when we evaluated the effects of CaCl2 concentration. The concentrations of CaCl2  was 0.2 mg/mL and CS was 0.4 mg/mL when we evaluated the effects of SA concentration. The concentrations of CaCl2 was 0.2 mg/mL and SA was 0.6 mg/mL when we evaluated the effects of CS concentration.

Reviewer 4 Report

Comments and Suggestions for Authors

Review of Foods-2858480

Modulation the Synergistic Effect of Chitosan-Sodium Alginate Nanoparticles with Ca2+: Enhancing the Stability of Pickering Emulsion on D-limonene

Overall assessment

This work studies a well know emulsion system based on chitosan/sodium alginate and evaluates the influence of Ca2+ over some properties of the nanoparticles and the emulsified oil. Although the manuscript is clear and well written, novelty is questionable. Below some recommendations for the authors:

Specific remarks/comments

1.       Introduction. Lines 82-99. Please include a better and deeper explanation about the novelty of the manuscript and what is new that it contributes with respect to the already published articles.

2.       Lines 102-107. Include the analytical grade or the reactants.

3.       Line 134. Why did you use 7% of D-limonene and 3% MCT. Is it optimized?

4.       Lines 134-140. Did you control the temperature during the preparation of the PE?

5.       Line 146. Define emulsion index or creaming index but use always the same. Did you control the temperature of the centrifugation? Were the emulsions prepared at the same moment or in advance?

6.       Line 209. Why did you perform the heating at 90 C? I think it could be excessive. Justify.  

7.       Line 210. What is the wavelength of the UV (A, B, or C) and how id the light irradiated. Please, include a measure to quantify the intensity of UV light exposition.  

8.       Figure 1. For each group (A, D, H) (B, E, I) and (C, F, J). What in the concentration of the other components (CaCl2, SA, CS).

9.       Figure 2. FTIR graph it is impossible to read properly. Size is very small.

10.   Figure 4 and 6. Key to the SEM or confocal images are impossible to read, make the figures bigger, includes some notes on the images or include the information in the captions.

11.   Do you have pictures of the PEs to include in the manuscript? Or even before and after the treatment with UV and heating.

Author Response

Dear reviewers,
thank you very much for taking the time to review this article. Based on your suggestion, I made the following modifications:

Comment 1: Introduction. Lines 82-99. Please explain the novelty of the manuscript better and in more depth, and what new contributions it makes to the already published article.

Response 1: Thank you very much for your suggestion. We have made the following changes to lines 79-94: Therefore, in this study, a stable D-limonene PE was prepared by attaching chi-tosan-sodium alginate nanoparticles (CS-SA/Ca2+ NPs) to the surface of D-limonene droplets.  The effects of CaCl2 concentration, SA concentration, CS viscosity and con-centration on particle size, PDI, and potential, as well as the effects of CS-SA/Ca2+ NPs concentration and homogeneous speed on the droplet size, zeta potential, and encap-sulation efficiency of D-limonene were investigated.  The morphology of CS-SA/Ca2+ NPs and CS-SA NPs, the microstructure, antioxidant and antibacterial properties of PE-coated D-limonene were also investigated.  Furthermore, the antioxidant and anti-bacterial activity of free D-limonene and D-limonene PEs were further compared after heating or UV irradiation.  This study provided a helpful reference for the application of CS-SA NPs-stabilized PE, and improved the stability of D-limonene essential oil, which was conducive to expanding the application of D-limonene in food industry.

Comment 2: Lines 102-107. Analytical grade or reactants are included.

Response 2: Thanks for the correction, we added the analytical grade of the reactant to line 97-102.

Comment 3: Line 134. Why use 7% D-limonene and 3% MCT.Is it optimized?

Response 3: Yes. The results showed that when the ratio of water phase to oil was 10:1 and the ratio of D-limonene to MCT in oil phase was 7:3, the particle size of the emulsion was smaller.

Comment 4: Lines 134-140. Did you control the temperature during the preparation of the PE?

Response 4: The emulsion preparation process is always carried out under ice bath conditions. (lines 130-136).

Comment 5: Line 146. Define an emulsion index or emulsion index, but always use the same method. Do you control the temperature of the centrifugation? Was the emulsion prepared at the same moment or beforehand?

Response 5: The cream index has been defined. Emulsions of the same factor are prepared at the same time, and the emulsions prepared in advance may cause certain experimental errors. Centrifugation of the emulsion is performed at 4 °C (rows 144-146).

Comment 6: Line 209. Why is heating at 90°C? I think this may be excessive. Plead.

Response 6: Because the emulsion is expected to be used in food cling film. In the production process of cling film, the temperature is high, and some even reach more than 100 °C. If the emulsion is used in edible cling film, it will produce high temperature, when the film is heated with food, so we chose 90 °C as the experimental condition.

Comment 7: Line 210. What is the wavelength of ultraviolet light (A, B, or C) and how the light is irradiated. Please include a measure to quantify the intensity of UV exposure.

Response 7: The samples were added into transparent glass bottles with the diameter of 1.3 cm and irradiated under UV light at 395 nm (UVA) in a camera obscura for 8 h. (line 210-214).

Comment 8: Figure 1.For each group (A, D, H) (B, E, I) and (C, F, J). What are the concentrations of the other components (CaCl2, SA, CS).

Response 8: When we assessed the effect of CaCl2 concentrations, the concentration of SA was 0.6 mg/mL and that of CS was 0.4 mg/mL. When we assessed the effect of SA concentrations, the concentrations of CaCl2 were 0.2 mg/mL and CS were 0.4 mg/mL. When we assessed the effect of CS concentrations, the concentration of CaCl2 was 0.2 mg/mL and SA was 0.6 mg/mL. (lines 257-261).

Comment 9: Figure 2. The FTIR graph cannot be read correctly. The size is very small.

Response 9: We zoomed in on the size of Figure 2C (FTIR plot). (line 115).

Comment 10: Figure 4 and 6.The key to an SEM or confocal image is unreadable, to make the number larger, to include some annotations on the image or to include information in the caption.

Response 10: We added some annotations to the SEM and CLSM images to make the numbers clearer. (line 361,454).

Comment 11: Do you have a photo of PE to include in the manuscript? Even before and after UV and heat treatment.

Response 11: Thank you for your guidance. We used to have these photos, but we lost them due to a phone malfunction. I am sorry that I am not in school due to the holiday, so I cannot provide it for the time being. If the article needs later, we can provide it after February 15th.

Round 2

Reviewer 3 Report

Comments and Suggestions for Authors

Accept in present form.

Reviewer 4 Report

Comments and Suggestions for Authors

The authors have addressed most of the comments and recomendations correctly.